# Uneven Distribution of Urban Green Spaces in Relation to Marginalization in Mexico City

**Cristina Ayala-Azcarraga** [1], **Daniel Diaz** [2], **Tania Fernandez** [1], **Fernando Cordova-Tapia** [3] and **Luis Zambrano** [1,*]

1 Instituto de Biología, Universidad Nacional Autónoma de México, Ciudad de México 04510, Mexico; cristina.ayala@fa.unam.mx (C.A.-A.); tania.fernandez@st.ib.unam.mx (T.F.)
2 Facultad de Ciencias, Universidad Nacional Autónoma de México, Ciudad de México 04510, Mexico; ddiaz@ciencias.unam.mx
3 Instituto de Ciencias del Mar y Limnología, Universidad Nacional Autónoma de México, Ciudad de México 04510, Mexico; fcordova@cmarl.unam.mx
* Correspondence: zambrano@ib.unam.mx

**Abstract:** The present study examines the spatial distribution and level of accessibility of urban green spaces (UGSs) within the context of Mexico City, with a particular focus on their relationship with marginalization. The study examined five distinct categories of UGSs based on their size and subsequently analyzed their total surface area per capita in relation to their correlation with the marginalization index. The data were subjected to descriptive statistical analysis, and correlations were computed to investigate the relationships between variables. We found 1353 UGSs accessible for public use with a total area of 2643 ha. Seventy-four percent of them had <1 ha of surface area, and 51% were located in only three municipalities that were mostly middle- and high-income. These municipalities concentrated a higher area of green spaces per capita. We found a negative correlation between the marginality index and the area of UGSs per municipality; the lower the marginality index was, the higher the area of green spaces. These results suggest that a bad distribution of UGSs can increase environmental injustice since urban environmental services are unequally distributed, affecting particularly marginalized populations. This research is a valuable contribution to the existing body of knowledge regarding the accessibility of UGSs in Mexico City, particularly in connection to marginalized communities. It emphasizes the significance of this topic in the context of environmental justice, urban sustainability, and the formulation of urban policy decisions. By engaging with these concerns, individuals can strive to foster a city that promotes fairness and well-being for all of its residents.

**Keywords:** urban green spaces; urban nature; environmental justice; well-being





## 1. Introduction

Large human migration to cities in the last century is due to employment opportunities and better access to urban areas' services. Consequently, more than 50% of the world's population live in cities, and the percentage will grow rapidly in the next two decades [1]. However, the process of urbanization has had an impact on the well-being of the people living in urban areas, generating contradictions. For example, living in large cities provides inhabitants with better access to health services. At the same time, urban areas are polluted and promote exposure to factors that make their population more vulnerable to diseases [2–4]. This scenario has increased interest in understanding which variables within cities influence human well-being, especially under the vision of sustainability.

One of these variables is the urban green spaces (UGSs) concept, which is increasingly essential in city studies. There is growing evidence of the ecosystem services that nature provides to cities, which are fundamental to people's well-being and quality of life [5–8].

For example, UGSs sequester carbon, mitigate air pollution, influence infiltration capacity, preserve biodiversity, and buffer temperature in cities [9–13]. From a social point of view, the presence of UGSs plays a fundamental role in establishing a support network by enhancing social cohesion within a community [14–17]. This role facilitates interactions between diverse social groups associated with local economic improvement [18,19] and strengthening local security [10,20,21]. At the individual level, the consequences of exposure to natural environments for recreational purposes are evident through the prevalence of mental and physical aids [22–26]. A lack of access to these spaces contributes to a greater susceptibility to pathologies observed more frequently among urban populations than rural ones [27]. Vegetation promotes physical activity, mitigates stress levels, and contributes to mitigating mental exhaustion, depression, decreased productivity, reduced irritability, and the potential for aggressive or hostile actions [22,28]. Contact with nature can improve cognitive and emotional abilities [29,30]. Even a brief 15 min exposure facilitates introspection and contemplation [31].

Additionally, USGs reduce the unequal vulnerability of some segments of society to extreme events [13,32,33]. These multiple benefits reported during the last decade have promoted UGSs to be at the center of the scientific discourse on urban sustainability [5,6]. Part of this discourse relates to the spatial distribution of UGSs since their benefits to the urban community suggest that the equitable spatial distribution of USGs improves social and individual well-being [34].

However, both UGS distribution and accessibility within cities tend to be uneven. The asymmetric distribution of these spaces may have historical origins in urban development. Nevertheless, this inequality may also be related to socioeconomic indicators such as the income, education, and ethnicity of the residents [35]. Without an equitable distribution of UGSs, a disparity in benefits can be generated among individuals. This asymmetry in access to green areas has been recognized as a problem of environmental injustice.

To understand the relationship between environmental justice and the distribution of green spaces, we use Mexico City as a case study. This city is tremendously unequal and has a significant number of green areas in its territory that are very heterogeneously distributed. First, it is necessary to analyze the concept of "green spaces" that are quantified in the official inventories of this city. Since the idea of UGSs has developed over time, different classifications vary depending on criteria such as use, size, or ownership regime.

In Mexico City, the Environmental Law defines the UGS concept as "All areas covered with natural or induced vegetation located in Mexico City". This definition includes any wooded area from an individual to a protected natural area of thousands of hectares [36]. With such a broad definition, each local governmental institution redefines a green space according to its own criteria, resulting in different official inventories [37]. Additionally, this definition does not consider characteristics such as size or accessibility, which differentially affect well-being. Since the middle of the last century, human use has been considered a primary criterion. Public or private spaces offer security to users, optimal conditions for the practice of sports or games and walks, and moments of recreation and rest, in which the fundamental compositional element is vegetation [38].

We have previously found that the specific characteristics of UGSs in Mexico City can differentially affect their users' well-being by influencing how they relate to green spaces [39]. Such results emphasize the importance of a better understanding of Mexico City's UGSs. However, for this to be reflected in the well-being of the inhabitants of Mexico City, it is necessary to realize a geography that can incorporate people's interaction with these spaces, considering their size and usability as determining characteristics. Therefore, the definition most satisfactory to the goals of the present work should include accessibility, in which case we used "public space that has vegetation and offers opportunities for recreational activities and moments of rest". This definition encompasses the usability of green spaces as a criterion for analyzing their effect on society.

This paper aims to provide an analysis of the distribution, number, size, surface, and spatial accessibility of UGSs in Mexico City and to evaluate these characteristics under

different levels of social marginalization. We hypothesize that the accessibility of green spaces is associated with spatial dynamics and socioeconomic variables.

## 2. Materials and Methods

### 2.1. Study Site

Mexico City has 9.2 million inhabitants in an area of 1485 km$^2$ (148,500 ha) and, from a political point of view, is divided into 16 municipalities; its total area is divided into urban and conservation zones (Figure 1). Indeed, the space assigned for conservation represents 58% (58,000 ha) of the total area of the city [17]. Nevertheless, in Mexico City, the use of conservation areas is restricted; therefore, conservation areas, private gardens, and inaccessible greenness were excluded from the concept of UGSs defined in this study. In this sense, one of the 16 municipalities (Milpa Alta) was intentionally excluded from the study since its entire territory is considered a conservation area and thus entails a lack of data for the size, distribution, and characteristics of UGSs within its territory.

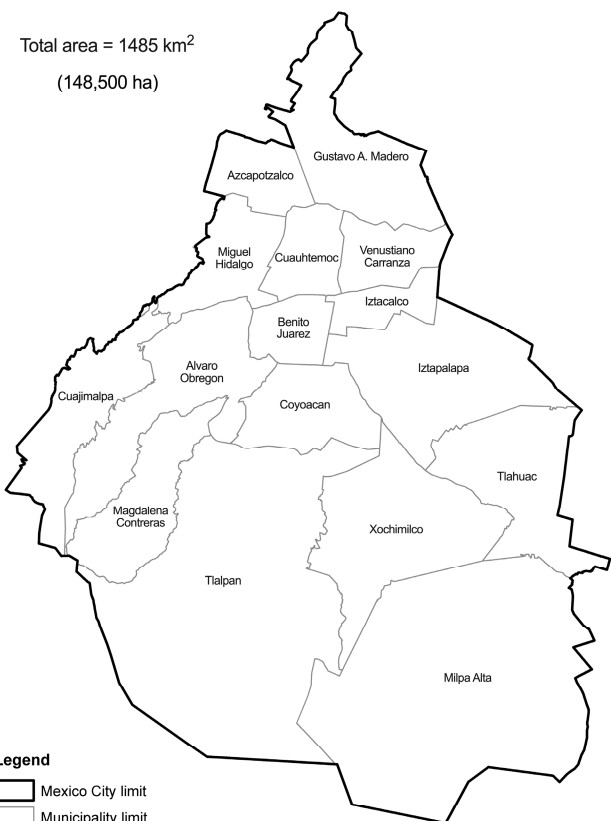

**Figure 1.** Total area and spatial distribution of the 16 municipalities of Mexico City.

### 2.2. Classification of Urban Green Spaces

For the characterization of UGSs in Mexico City, we considered five categories of green areas based on size as the main criterion, which is the same as the information of the urban cartography elaborated by the National Institute of Statistics and Geography [40]. From this cartographic information, we selected green spaces, parks, public gardens, wide ridges (with central passage), and sports facilities and classified them according to the following categories of UGSs: ridges (<0.5 ha); 1 ha (>0.5 and <1 ha); 1–5 ha (>1 and <5 ha); 5–10 ha (>5 and <10 ha); and 10–40 ha (>10 and <40 ha). In the case of ridges, only those measuring >20 m (wide) were included because they usually have a central walkway that allows for the use of these sites.

We selected such categories because they represent a gradient of sizes that allows for the realization of different activities that would potentially have a social impact on the community. The delimitation of each size category was based on a modification of the

classification of UGSs proposed by Ballester-Olmos and Morata [41], which considers that for each of these classes, there are specific characteristics that allow us to distinguish among them. After conducting a first characterization of the UGSs for this study, we eliminated those UGSs that, because of their geographical location, cannot be considered public or used. This ensured that the UGSs considered in this study were susceptible to being used by any inhabitants of Mexico City.

### 2.3. Spatial Distribution, Size, and Number of Urban Green Spaces among Municipalities

We elaborated a basic characterization of the UGSs of Mexico City in ArcGIS 10.1 using the existing cartographic information of urban localities and trees [42]. ArcGIS is a complete system for collecting, organizing, managing, analyzing, sharing, and distributing geographic information. It consists of a series of tools that allow you to perform professional GIS work. From these data, we obtained: (1) the number and total surface area of UGSs per size category and (2) the number and surface area of UGSs by size category per municipality. The area of UGSs per capita was also obtained, considering the number of people for each municipality according to the published data of the census 2014 (6.2 version) conducted by the National Institute of Statistics and Geography [40].

### 2.4. Association of Urban Green Spaces and Socioeconomic Indicators

In addition, to examine a possible association between the spatial distribution and characteristics of the UGSs and the marginalization level of the municipalities, we included the marginality index [43]. According to the National Population Council (CONAPO), marginalization is a multidimensional and structural phenomenon that is expressed in the lack of opportunities and the unequal distribution of progress in the productive structure, influencing levels of well-being and capacity-building resources and therefore in development. The marginality index is a summary measure that allows differentiating the different geographical units according to the global impact of the deficiencies suffered by the population and can be used as a proxy of socioeconomic development (Table 1).

**Table 1.** Socioeconomical indicators of the marginalization index [43].

| Socioeconomical Dimensions | Ways of Exclusion |
| --- | --- |
| Education | Illiteracy<br>Population without elementary school |
| Dwelling | Private dwellings without drainage or sanitary service<br>Private dwellings without electricity<br>Private dwellings without piped water<br>Private dwellings with some level of overcrowding<br>Private dwellings with dirt floor |
| Population distribution | Localities with less than 5000 inhabitants |
| Monetary income | Occupied population that receives up to two minimum wages |

### 2.5. Statistical Analysis

We used descriptive statistics to quantitatively characterize and compare the composition of UGSs in the municipalities according to size categories. One-tailed Pearson simple correlation analyses were used to assess the relationship between the area per capita of UGSs and the population density and marginality index of the municipalities. Additionally, to further examine the pattern of distribution of the UGSs within Mexico City, we first analyzed the percentile of distribution for each size category, and then we integrated each size category per municipality with a cluster analysis to show similarities between localities. Additionally, we used a heatmap based on the Z score to visualize the similarities in maps as described elsewhere [44,45]. Except for the cluster analysis that was performed with SAS OnDemand for Academics (SAS Institute, Cary, NC, USA), all graphs and analyses were performed on Prism 10 (GraphPad Inc., San Diego, CA, USA).

## 3. Results

### 3.1. Number and Surface Area of Urban Green Spaces in the Municipalities of Mexico City

Mexico City has 2096 neighborhoods, of which 450 have restricted access, so their green spaces are not open to all the inhabitants of the city. We found 1353 UGSs with a total area of 2643 ha that represents 1.8% of the total area of Mexico City. These 1353 UGSs coincided with the definition proposed in this study and were distributed as follows: ridges, 40.1% (543); 1 ha, 33.9% (458); 1–5 ha, 15.6% (211); 5–10 ha, 3.5% (48); and 10–40 ha, 6.9% (93) (Table 2). According to the size distribution, the smaller the size of the UGSs, the greater their abundance; furthermore, in Mexico City, 74% of the UGSs are <1 ha in size. The 1353 UGSs represented a total of 2643 hectares, of which 16.0% and 53.3% were found in the ridges (424 hectares) and 10–40 ha in UGSs (1408 hectares), respectively. The remaining three groups contributed together with 30.7% of the total extension of UGSs in Mexico City: 1 ha, 167 hectares; 1–5 ha, 377 hectares; and 5–10 ha, 267 hectares.

**Table 2.** Number of UGSs per size category and total surface for each municipality.

| Municipality | Categories of UGSs | | | | | Total Number of UGSs | Total Surface (ha) |
|---|---|---|---|---|---|---|---|
| | Ridge | 1 ha | 1–5 ha | 5–10 ha | 10–40 ha | | |
| Coyoacan | 53 | 171 | 37 | 4 | 6 | 271 | 602 |
| M. Hidalgo | 23 | 71 | 56 | 19 | 51 | 220 | 561 |
| Gustavo A. M. | 170 | 16 | 6 | 5 | 7 | 204 | 472 |
| Tlalpan | 34 | 49 | 22 | 6 | 10 | 121 | 399 |
| V. Carranza | 28 | 23 | 9 | 3 | 8 | 71 | 139 |
| A. Obregon | 47 | 25 | 32 | 2 | 5 | 111 | 122 |
| Iztapalapa | 102 | 37 | 8 | 1 | 0 | 148 | 92 |
| Azcapotzalco | 17 | 11 | 6 | 0 | 2 | 36 | 61 |
| Cuauhtemoc | 23 | 24 | 14 | 1 | 0 | 62 | 55 |
| Iztacalco | 35 | 8 | 4 | 1 | 0 | 48 | 45 |
| B. Juarez | 4 | 18 | 11 | 3 | 0 | 36 | 41 |
| M. Contreras | 2 | 5 | 3 | 2 | 4 | 16 | 38 |
| Tlahuac | 0 | 0 | 0 | 1 | 0 | 1 | 9 |
| Xochimilco | 5 | 0 | 0 | 0 | 0 | 5 | 3 |
| Cuajimalpa | 0 | 0 | 3 | 0 | 0 | 3 | 3 |
| Total | 543 | 458 | 211 | 48 | 93 | 1353 | 2643 |

Across the 15 municipalities of Mexico City, the number of UGSs ranged broadly between 1 and 271. Three municipalities (Coyoacan, M. Hidalgo, and Gustavo A. Madero) accounted for 51.4% of the total number of UGSs (695/1353) found in Mexico City. In clear contrast, municipalities such as Xochimilco, Cuajimalpa, and Tlahuac in conjunction had nine UGSs that represented only 6.6% of the total. Because of this heterogeneous distribution of the number of UGSs, the total surface covered by all the size categories within each municipality also exhibited contrasting patterns (Table 2). In municipalities such as Coyoacan, Gustavo A. Madero, and M. Hidalgo, the extension of UGSs ranged from 472 to 602 hectares and accounted for 61.6% of the total surface of 2643 hectares found in Mexico City. For the three municipalities with the lowest number of UGSs, their total surface green areas were 15 hectares, which represented only 0.6% of the total. These results emphasize the disparities among municipalities regarding the distribution and surface covered by the different categories of UGSs.

### 3.2. Composition of the Green Spaces among Municipalities of Mexico City

As depicted in Figure 2b, four municipalities showed a total surface >375 hectares of their UGSs. In these four municipalities, UGSs of 10–40 ha contributed the highest amount of surface: Gustavo A. Madero, 253 hectares (53.6%); Tlalpan, 313 hectares (78.4%); M. Hidalgo, 328 hectares (58.6%); and Coyocan, 352 hectares (58.5%). In contrast, ridges contributed the greatest amount of UGS surfaces in places such as Iztapalapa (59 hectares, 64.1%), Iztacalco (27 hectares, 60%), and Xochimilco (3 hectares, 100%).

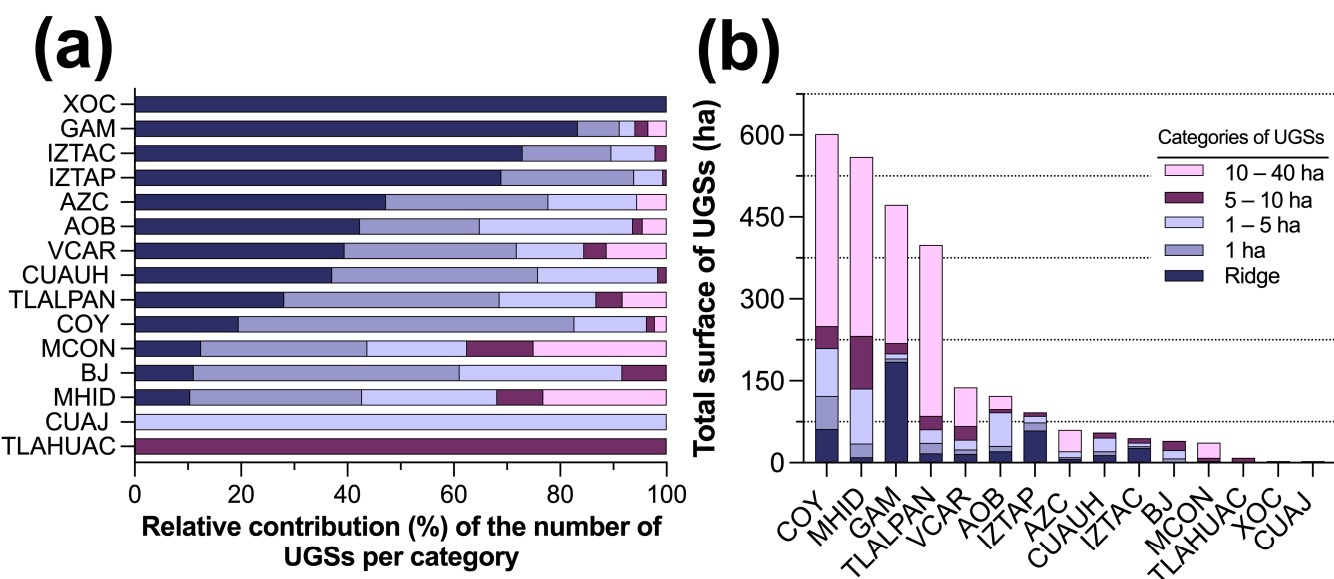

**Figure 2.** (**a**) Proportion of each size category of UGSs per municipality. COY: Coyoacan; MHID: Miguel Hi-dalgo; GAM: Gustavo A. Madero; VCAR: Venustiano Carranza; AOB: Alvaro Obregon; IZTAP: Iztapalapa; AZC: Azcapotzalco; CUAUH: Cuauhtemoc; IZTAC: Iztacalco; BJ: Benito Juarez; MCON: Magdalena Contreras; XOC: Xochimilco; CUAJ: Cuajimalpa, and (**b**) total surface (Ha) of the UGSs for each municipality.

In each municipality, the relative contribution of each size category of the UGSs showed an interesting pattern, according to which, there were both a disparate distribution of the UGSs across municipalities and a heterogeneous composition of the green spaces within each municipality (Figure 2). For instance, Xochimilco, Cuajimalpa, and Tlahuac presented only one category of UGSs, while in contrast, five and seven municipalities had four to five size categories, respectively. Except for Tlahuac, the relative contribution of UGSs with a size of 5–10 ha was the lowest in the municipalities (2.21–25%). In M. Hidalgo and M. Contreras, the contribution of ridges was lower (10.4 and 12.5%), while UGSs of larger size were more frequent. Indeed, in these two municipalities, the percentage of UGSs 10–40 ha was the greatest among all locations (25.0 and 23.18%, respectively).

### 3.3. Spatial Distribution of UGSs across the Municipalities of Mexico City

To illustrate the pattern of the distribution of UGSs across the municipalities of Mexico City, in Figure 3a, we depict the spatial distribution of all the green areas mapped in our study. As judged by the accumulation of a higher number of green spaces in some locations, there is a characteristic uneven distribution in both the number and surface area of UGSs among the municipalities of Mexico City. Furthermore, to show these striking differences at a higher resolution, in Figure 3b,c, we present two municipalities (M. Hidalgo and M. Contreras) as representatives of the inequity of the distribution of UGSs in Mexico City. According to Figure 2a, these two municipalities showed similar UGS compositions. Nevertheless, the number and size of the green spaces mapped in each locality varied broadly between them.

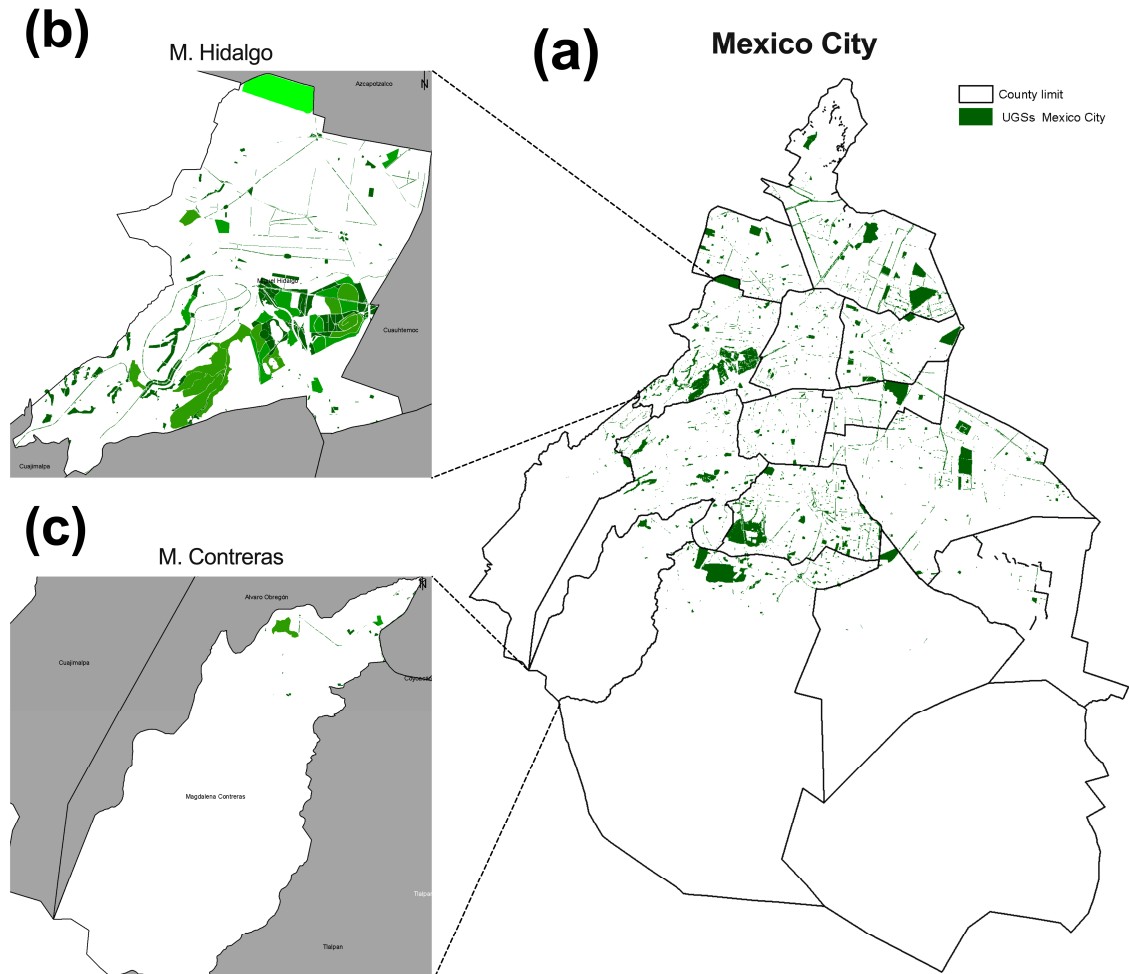

**Figure 3.** (**a**) Spatial distribution of the UGSs in Mexico City and mapping of the five categories of UGSs in (**b**) M. Hidalgo and (**c**) M. Contreras municipalities, which were selected to show the disparate pattern of distribution.

The maps shown in Figure 4a illustrate the heterogeneous spatial distribution of the percentiles of UGSs among the municipalities of Mexico City. Despite the contrasting pattern found across size categories, the adjacent municipalities of Xochimilco and Tláhuac were consistently distributed into the lowest percentile category (<16.6 th). Likewise, a similar distribution was found for Cuajimalpa. In contrast, the percentile distributions for the UGSs of Tlalpan, Coyoacan, and M. Hidalgo consistently had higher values. As shown in Figure 4b, the integrated analysis of the size categories of UGSs revealed the formation of two distinct groups of municipalities in Mexico City: those with below-average values (negative Z scores and green-colored) and municipalities with above-average scores (positive Z scores and red-colored).

Nine municipalities that were characterized by a reduced number of UGSs for each category formed a well-defined cluster in the upper portion of the dendrogram. In the middle section were Tlalpan and A. Obregon, which were grouped into a single cluster due to their increased number of UGSs. Finally, Iztapalapa, Gustavo A. Madero, Coyoacan, and M. Hidalgo, all of which were characterized by above-average values for at least one of the size categories of UGSs, were separated into four distinct clusters in the dendrogram.

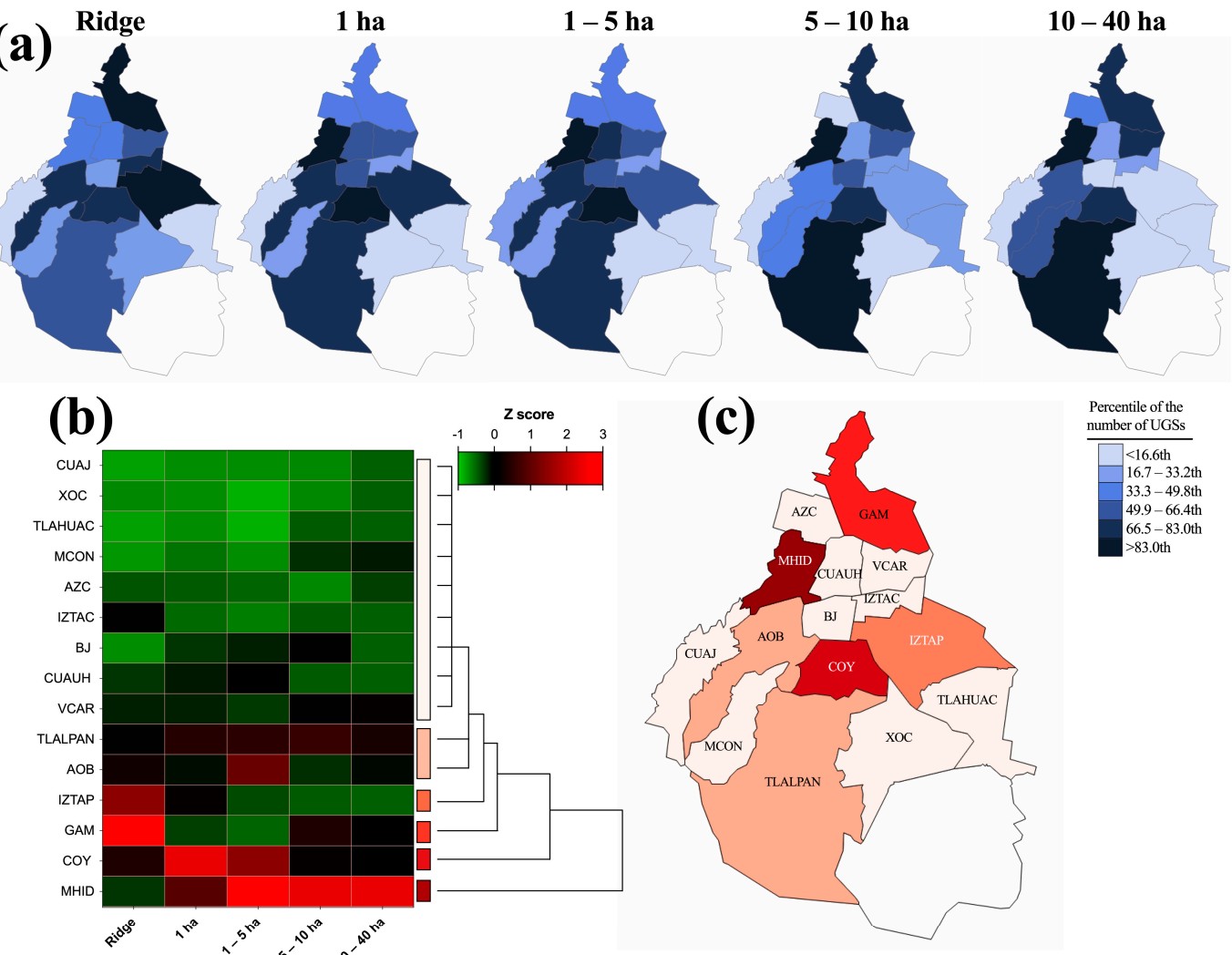

**Figure 4.** (**a**) Spatial distribution of the percentile per size category of the UGSs among the municipalities of Mexico City, (**b**) heatmap with cluster analysis of the Z score of the number of UGSs, and (**c**) spatial representation of the clusters.

Furthermore, as shown in the spatial representation of the multivariate analysis (Figure 4c), four of the municipalities grouped into one of the clusters were indeed adjacent localities (Cuauhtemoc, V. Carranza, B. Juarez, and Iztacalco). In addition, a central band of three municipalities that included A. Obregon, Coyoacan, and Iztapalapa showed an increased number of UGSs. Interestingly, the three municipalities with the highest Z scores (Gustavo A. Madero, M. Hidalgo, and Coyocan) were heterogeneously distributed across the middle-upper portion of Mexico City.

### 3.4. Association between UGS Socioeconomic Indicators in Mexico City

There was a differential distribution of the UGSs that varied according to the marginalization level of the localities (Figure 5a). We found a negative correlation (r = −0.47, $p$ = 0.037) between the marginality index and the area of UGSs per municipality; the lower the marginality index was, the higher the area of green spaces (Figure 5b). We also observed an unequal distribution of the area per capita of green spaces among municipalities (Figure 5b).

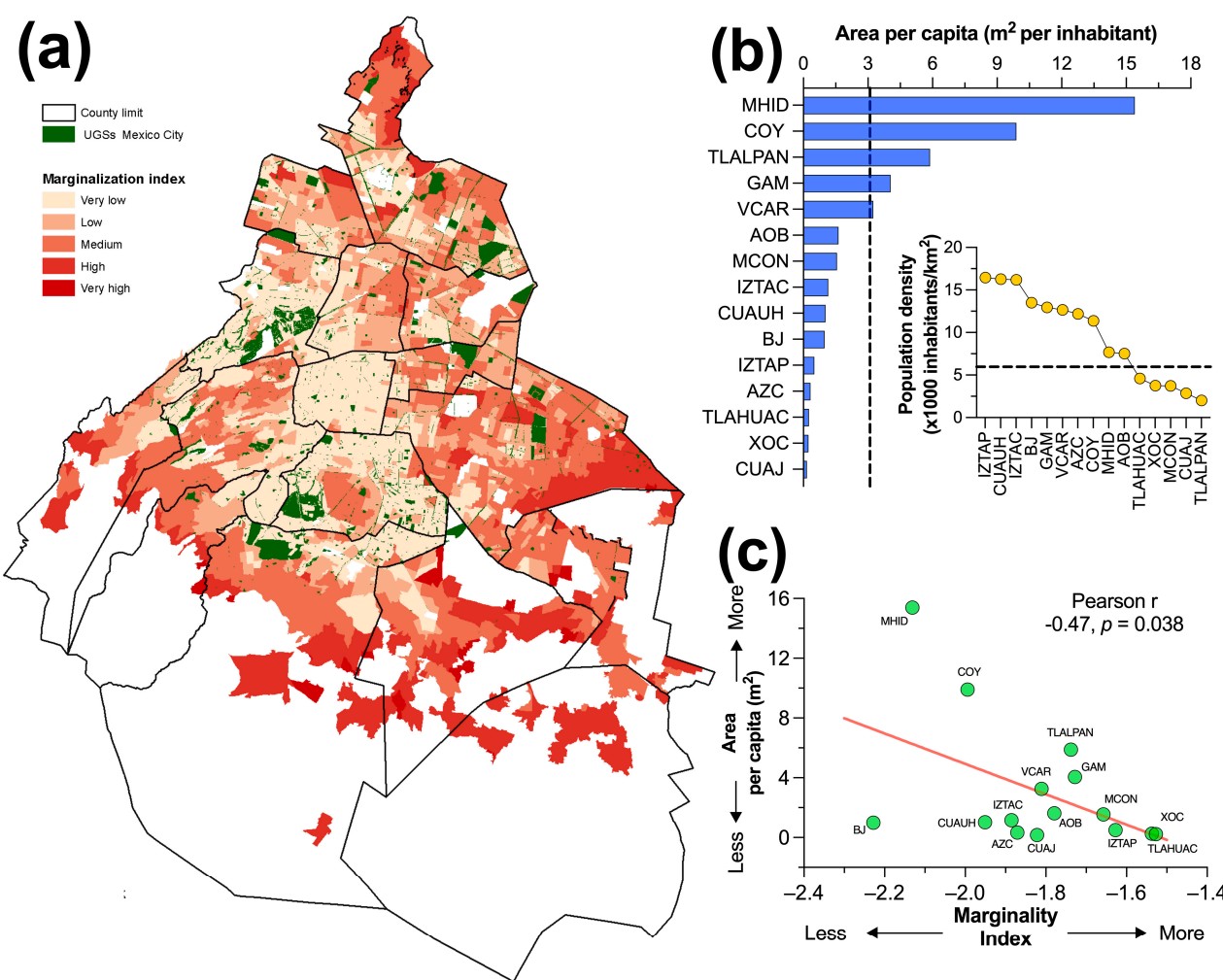

**Figure 5.** (**a**) Merge of the spatial distribution of UGSs in Mexico City and the marginality index for the localities of each municipality, (**b**) comparison of the area per capita of UGSs, and (**c**) association between the socioeconomic indices and the area per capita of UGSs for the municipal-ities of Mexico City. In **b**, the dotted lines refer to average values for Mexico City. In **c**, the red line represents the linear trend which was added to assist interpretation of the linear association.

From the mapped UGSs, we found an average value of 3.1 m² of green spaces per inhabitant in Mexico City. However, there was a broad discrepancy in the values among localities; for instance, M. Hidalgo has 15.4 m² per inhabitant, followed by Coyoacan with 9.9 m², whereas marginal municipalities such as Iztapalapa, Azcapotzalco, Tlahuac, Xochimilco, and Cuajimalpa have less than 0.5 m² of green spaces per inhabitant. Further-more, we hypothesized that the area per capita of green spaces should be related to the population density of each municipality, although we did not find a correlation between these variables (r = −0.08, *p* = 0.769). This result indicated that in those municipalities with the highest population density per km², there is a lack of a corresponding high value of green spaces per m²; thus, UGSs might not be enough for the inhabitants of the more popu-lated locations. Interestingly, there were two municipalities (B. Juarez and Cuauhtemoc) that, despite their lower values of marginality, did not have a corresponding high per capita area of green spaces, as their values were 0.98 and 1.02 m² per capita, respectively.

## 4. Discussion

The accessibility and size of UGSs play a key role in their usability and benefits to urban citizens [13,46,47]. This research provides information on the relationship between these characteristics and environmental justice in cities using the case of Mexico City. Our

results suggest an asymmetric distribution and a predominance of small UGSs in Mexico City, which does not differ from the international trend [48–53]. By comparing the results with other cases in Mexico, we also found similarities in the unequal UGS distribution. Previously, Fernandez-Álvarez reported an inverse relationship between public parks in Mexico City and poverty data [54]. Our results also coincide with other Mexican cities, such as San Luis Potosí and León, where there is also an inequitable relationship between the distribution of UGSs and economic income, favoring the high-income region [55,56]. The more significant amount of smaller green spaces may limit the potential benefits for residents' mental health and physical activity [32,57]. Previous research has shown that access to larger UGSs is associated with significant improvements in mental health, highlighting the importance of addressing the size-related distribution of UGSs [58,59].

The asymmetry in the distribution of UGS in Mexico City [60] is reflected in an inversely proportional relationship between the level of marginalization and access to these spaces. This situation, where the most marginalized municipalities have fewer UGSs and are smaller, suggests environmental injustice. Marginalized populations are precisely those most affected by diseases related to sedentary lifestyles and, in turn, are linked to a lack of access to UGS. Our results can be related to the report provided in 2021 by the Health Ministry of Mexico City (SEDESA), where 71.4% of the 281,638 people who attended a health program in Mexico City presented comorbidities [61]. In this report, Iztapalapa municipality presented the highest prevalence of overweight (17.3%), obesity (16.3%), diabetes (13.1%), and hypertension (8.7%), which is also the municipality that we found with the lowest number of UGSs per capita with less than 30 cm per person. In contrast, the municipality with the highest UGSs per capita of 14.5 $m^2$ per person (Miguel Hidalgo) reported incidence rates of less than 2.5% for all the diseases mentioned. Even with such a clear relationship, the primary purpose of this study was not to establish a causal association between diseases linked to sedentary lifestyles and the availability of UGSs. Therefore, future research should focus on examining whether there is a correlation between these factors.

The scientific evidence surrounding the benefits of green spaces for physical and mental health is strong, underscoring the urgency of integrating this research into UGS planning and management. Given that the inequitable distribution of UGSs disproportionately affects marginalized populations, addressing this issue is essential to reduce inequity and promote the well-being of all inhabitants. The results also enhance the importance of the diversity of UGS sizes in the social context. Larger green spaces foster social cohesion by allowing diverse activities simultaneously [62]. However, the predominance of smaller UGSs may limit interactions between different groups, which has implications for forming cohesive communities. Future studies should examine how the lack of size diversity may influence the perception of safety and social cohesion in colonies with different social strata. The results also highlight the need to consider watershed and local scales in UGS management. While green spaces at the watershed level provide ecosystem services to the entire urban area, local UGSs directly impact the inhabitants' quality of life [63]. UGS management and planning should consider these two scales to address the problems more effectively and fairly.

At the local scale, benefits are directly related to the accessibility of that particular UGS. Although environmental justice advocates an equitable distribution of green space per capita, distribution by itself does not guarantee universal access to UGSs for a city's residents. It is crucial to consider variables such as a safe and easily accessible environment for all demographic groups, including people with disabilities, limited mobility, and newborns. According to Jennings et al. (2016), several factors can potentially discourage the use of UGSs [64]. These factors include the perception of crime, challenges associated with accessing UGSs due to their location in areas with high levels of automobile traffic, undesirable wildlife such as rodents, and the maintenance of vegetation in the vicinity [65]. Therefore, to achieve proper UGS accessibility, it is imperative to discern the quality and

functional attributes within the spatial context, such as those found in the surrounding urban matrix [6,13,66].

UGSs within road infrastructure or parks in the middle of two highways have poor accessibility and should not be considered for use. However, this is not always the case, as the accessibility and utilization of UGSs are not considered in official inventories. Our results indicate that when considering exclusively public green spaces with the potential to be used, excluding tree-lined roads that are inaccessible for public use, the inhabitants of Mexico City possess an area of green spaces five times lower than the figure reported by the Procuraduría Ambiental y del Ordenamiento Territorial (PAOT) in the 2015 census [37]. This overestimation of UGSs significantly impacts the imperative of implementing public policies to improve the conditions of UGSs.

Despite the robustness of the information obtained, we recognize that the quality and availability of the data used may influence the results. In addition, perceived safety, connectivity, and other factors can affect the actual accessibility of UGSs [67,68]. To fully understand the complex reasons behind these disparities and their impact on the community, future research should explore historical, urban, economic, and political aspects contributing to the unequal distribution of UGSs in Mexico City.

The findings of this study suggest that, as hypothesized, there is a significant association between UGS accessibility and the economic marginalization of social groups within Mexico City. These results underscore the importance of urban design that not only prioritizes the usability of green spaces for all city residents, but also prioritizes marginalized communities. Creating and improving UGSs in socioeconomically disadvantaged regions could be a fundamental approach to promoting ecological equity and enhancing the well-being of inhabitants [32]. Our results emphasize the need for urban planning and policy informed by research and environmental justice to address the unequal distribution of UGSs. Enhancing the well-being of residents can be achieved through creating and making UGSs more accessible in marginalized areas, as well as a comprehensive understanding of the factors contributing to this issue. Future studies should attempt to understand why these differences exist by analyzing the various historical, urban, social, economic, and political factors involved in the number and size of UGSs.

## 5. Conclusions

This research paper provides a comprehensive analysis of the spatial arrangement of UGSs in Mexico City, uncovering a notable disparity in both the availability and dimensions of UGSs. The observed discrepancy can be comprehended via the lens of the environmental justice paradigm, as it is intricately linked to socioeconomic variables such as degrees of marginalization, which hinder the ability to access these environments in regions with greater socioeconomic disadvantages. The presence of heterogeneity in the allocation of resources not only affects the welfare of individuals, but also represents a significant environmental injustice, as those who are most in need of the advantages provided by green places are often the ones with the least accessibility. In a similar vein, it is crucial to consider the constraints linked to the advantages provided by UGSs of various sizes. Smaller UGSs may possess a diminished capacity to provide benefits, particularly to persons who are susceptible to illnesses resulting from sedentary behaviors and elevated stress levels.

The provision of equitable access to green space is vital for the promotion of a healthy urban environment. The achievement of this objective can be attained through the prioritization of concepts pertaining to environmental justice, as well as the enhancement of the accessibility and scale of UGSs over the entirety of the city. To effectively address the matter of UGS accessibility and its implications for public health and well-being, it is imperative to embrace policies that are grounded in empirical research, as delineated in the present study. Based on the findings, it is advisable for public policy to prioritize neglected regions to effectively alleviate prevailing imbalances. In a similar vein, it is crucial to incorporate the requirements and viewpoints of marginalized communities to promote community

engagement and guarantee that policies effectively cater to the demands and preferences of the urban populace.

The advancement toward a more equitable and just urban environment can be enhanced by the implementation of urban planning strategies that prioritize the accessibility and functionality of UGSs while also demonstrating attentiveness to the needs and challenges faced by marginalized communities.

**Author Contributions:** Conceptualization, C.A.-A. and L.Z.; methodology, C.A.-A., T.F. and F.C.-T.; software and visualization, C.A.-A., T.F. and D.D.; formal analysis, C.A.-A. and D.D.; investigation and writing—review and editing, C.A.-A., T.F., F.C.-T. and L.Z.; data curation, C.A.-A. and T.F.; writing—original draft preparation, C.A.-A., T.F., D.D. and L.Z.; project administration, C.A.-A. and L.Z. All authors have read and agreed to the published version of the manuscript.

**Funding:** This research received no funding.

**Data Availability Statement:** All the datasets used to construct this study are available upon reasonable request to the corresponding author.

**Acknowledgments:** The first author gratefully acknowledges the PhD Program in Sustainability Sciences, UNAM (Posgrado en Ciencias de la Sostenibilidad, Universidad Nacional Autónoma de México). Additionally, we thank Ruth Luna Soria for her support during the preparation of the manuscript.

**Conflicts of Interest:** The authors declare no conflict of interest.

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
