# Peer review of "Uneven Distribution of Urban Green Spaces in Relation to Marginalization in Mexico City"

_sustainability, doi:10.3390/su151612652_

Round 1
Reviewer 1 Report
Abstract
The research method should be clearly stated.
The contributions of the study should be presented in this section.
Introduction
The research literature has not been fully reviewed. It is recommended to provide more complete research literature.
The theoretical framework and hypothesis for the study are missing and recommended to incorporate them to article.
Research gap and contribution of the study need to be fully explained.
Results
All the components of Figure 1 should be introduced.
The obtained results should be compared with other previous studies and discussed more fully.
What software was used for data analysis?
Conclusion
The caveats with approach should be added to the article.
This section is incomplete. It is necessary to provide policy applications based on research findings.
Reviewer 2 Report
This article mainly analyzes "what" , less about "why" and "how". It mainly studied the imbalance between the distribution and size of urban green space in Mexico, and its environmental injustice. In my opinion, in general, the biggest problem may be the Introduction part. The Materials and methods and Results are well written, including writing ideas and data visualization. Discussion has the same problem with Introduction, without combining with previous studies.
Abstract: Overall, good. I think it is better to add sentences that emphasize the significance of the research.
Keywords: I don't quite understand why the first keyword is urban parks instead of UGS.
Introduction: The concept of "UGS" is too long; no review of previous studies; the research ideas and methods of the last paragraph are too concise.
Line 53: ‘censed’ is misspelled, maybe it should be ‘censored’.
Materials and methods: Good. The five parts including research location, classification and reasons, data acquisition, index selection and data analysis method, are all carefully elaborated, with sufficient basis and worth learning and reference.
Line 117: ‘(0.5ha)’ maybe ‘(<0.5ha)’.
Results: Based on descriptive statistics, and the correlation analysis between per capita green area and marginalization index.The results are presented by tables, bar charts, heatmaps and cluster analysis. Data visualization forms are diverse and beautiful. The analysis is also comprehensive, and five types of urban green space are analyzed from the whole and among municipalities, which is logical and worth reading.
Line 169: Perhaps without tying the classification of <1ha with the classification of 10-40ha, directly highlighting 10-40ha accounts for more than 50%
Discussion: I think the main possible problem is not discussing in combination with previous studies.
Reviewer 3 Report
The present manuscript was submitted by five biological science members of the Mexican National University and presents a GIS-based study on urban green spaces of Mexico City of 15 of the 16 boroughs or demarcaciones territoriales and their social marginality index (income differences).
In their introduction, it should be stated clearly that Mexico City as a whole consists of 16 municipalities/boroughs/demarcaciones territoriales and 15 of them were studied. The reviewer, not familiar with Mexico City, had to look up how many boroughs the megacity actually has. So the study comprises the whole city with exception of one borough which is a conservation zone.
The study itself is explained and their results presented in an excellent way, easy to understand. Its main result, a negative correlation of social marginality/lower income with size and quality of urban green spaces, is not surprising and is found in other empirical studies on urban areas of the Americas (see their references).
The study under review cites 37 references. Of these, 3 are general thematic reviews (one including China), 10 deal with the present studies’ location, Mexico City, 2 of them involving identical authors. 7 quoted studies discuss US locations (3 Phoenix AZ; one each Baltimore, Kansas City, Milwaukee, and one 10 US cities), 1 is about Santiago, Chile, and 1 about Ontario, Canada. This means that the quoted location references are America-centered. What is missing are references to thematic local studies in Europe, Asia and Australia as a step to a worldwide generalizability of the green distributional injustice phenomenon.
This does not imply that the study under review would have to be re-designed, but results of its America-centered approach should be linked with some recent international studies. Here are six examples I found:
2022 Korpilo et al. (study location Copenhagen, Denmark)
Environmental justice in urban green and blue space planning
https://doi.org/10.1016/j.apgeog.2022.102794
2022 Kato-Huerta et al. (study location Las Palmas de Gran Canaria, Spain)
A distributive environmental justice index to support green space planning
in cities
https://doi.org/10.1016/j.landurbplan.2022.104592
2020 Mears et al. (study location Sheffield, England)
The case for using multiple indicators of neighbourhood greenspace
https://doi.org/10.1016/j.healthplace.2020.102284
2022 Pan et al. (study location London, England about COVID)
Evaluating the risk of accessing green spaces in COVID-19 pandemic: A
model for public urban green spaces (PUGS) in London
https://doi.org/10.1016/j.ufug.2022.127648
2022 Wu et al. (study about urban China locations)
Socioeconomic groups and their green spaces availability in urban areas of
China: A distributional justice perspective
https://doi.org/10.1016/j.envsci.2022.01.008
2022 Hsu et al. (study location four Australian cities)
Beyond the Backyard: GIS Analysis of Public Green Space Accessibility in
Australian Metropolitan Areas.
https://doi.org/10.3390/su14084694
By quoting and briefly linking with some studies from the rest of the world the main message of the study under review would not be confined to Latin America compared with the US and Canada but reach an international level, i.e. would show that the injustice/inequality phenomenon is not culture specific but a product of political-economic segregation of city populations that can be traced throughout the world, therefore in need of international attention (United Nations organizations).
Reviewer 4 Report
I believe that this work is extremely important for the study of cities and that, with this research, important steps are taken in understanding the need to locate green areas in cities regardless of the degree of marginalization of the population.
However the authors would need to pay attention to certain aspects that are necessary to improve the quality of the paper:
1. Please elaborate on the social marginality index, which is inserted in the abstract but not sufficiently explained during the paper. Is this the same with marginality index? Or marginalization index in Table 1?
2. Introduction of at least brief literature review section. A lot of work is being done on this very topic in MDPI journals. It could highlight what the gaps in the research are and how this work fits into the wider context of research.
3. Please outline more clearly what the objective of the paper is and why the research is novel.
4. The definition of UGS should be included in the section Materials and methods (lines 86-96). In addition, what is the classification of UGS in the literature? Moreover, you mention the hypothesis in line 257, please include it also in this section or in the Introduction. Further, please enter brief details of ArcGis 10.1 software.
5. Please keep a consistent style: Fig 2 (a) or 2A? 1b or 1B?
6. Please enrich the Conclusions section. Conclude if the Hypothesis has been confirmed
7. Please pay attention to the references - Reference 1, although an important work for the field, cannot be called an international guideline. It may be useful to consult official WHO, IUCN, US Green Building Council guidelines and standards, or other national documents.
Reviewer 5 Report
1. The topic selection of this paper has high social significance, reflecting the principle of people-oriented.
2. There are some good problems found in the article, which are not listed here.
3. It would be better if the author could further propose some corresponding solutions on the basis of the problem discovery.
4. In view of the uneven distribution of green space and environmental injustice in Mexico City, I hope the author can try to put forward some targeted solutions, including some urban planning methods, policies and systems.
5. In a word, the idea, analysis and conclusion of this paper are all good, but it is weak in the following solution strategies. I hope the author can make some supplements to further enhance the academic research value and practical guidance of this paper.
Overall, the English expression is smooth and can be further optimized.
Round 2
Reviewer 1 Report
The comments are fully addressed
Author Response
On behalf of my colleagues, we thank you for your consideration.
Reviewer 2 Report
The authors have responded all my concerns. I think it is acceptable for publication.
Author Response

(The authors gave the same response as above.)

Reviewer 4 Report
the authors have improved the quality of the paper, as suggested.
Author Response

(The authors gave the same response as above.)
